# FIND A WINNING SIGN:
# SIGN IS ALL WE NEED TO WIN THE LOTTERY

**Junghun Oh[1], Sungyong Baik[3,4], and Kyoung Mu Lee[1,2]**
[1]Dept. of ECE&ASRI, [2]IPAI, Seoul National University
[3]Dept. of Artificial Intelligence, [4]Dept. of Data Science, Hanyang University
{dh6dh,kyoungmu}@snu.ac.kr, dsybaik@hanyang.ac.kr

## ABSTRACT

The Lottery Ticket Hypothesis (LTH) posits the existence of a sparse subnetwork (a.k.a. winning ticket) that can generalize comparably to its over-parameterized counterpart when trained from scratch. The common approach to finding a winning ticket is to preserve the original strong generalization through Iterative Pruning (IP) and transfer information useful for achieving the learned generalization by applying the resulting sparse mask to an untrained network. However, existing IP methods still struggle to generalize their observations beyond ad-hoc initialization and small-scale architectures or datasets, or they bypass these challenges by applying their mask to trained weights instead of initialized ones. In this paper, we demonstrate that the parameter sign configuration plays a crucial role in conveying useful information for generalization to any randomly initialized network. Through linear mode connectivity analysis, we observe that a sparse network trained by an existing IP method can retain its basin of attraction if its parameter signs and normalization layer parameters are preserved. To take a step closer to finding a winning ticket, we alleviate the reliance on normalization layer parameters by preventing high error barriers along the linear path between the sparse network trained by our method and its counterpart with initialized normalization layer parameters. Interestingly, across various architectures and datasets, we observe that any randomly initialized network can be optimized to exhibit low error barriers along the linear path to the sparse network trained by our method by inheriting its sparsity and parameter sign information, potentially achieving performance comparable to the original. The code is available at https://github.com/JungHunOh/AWS_ICLR2025.git

## 1 INTRODUCTION

In the field of deep learning, over-parameterization is viewed as a key to enhancing network capacity and improving generalization (Neyshabur et al., 2019; Belkin et al., 2019). It is well known that after training an over-parameterized dense network, many redundant parameters arise that can be removed without affecting performance, leading to the emergence of network pruning techniques (Liu et al., 2017; Lin et al., 2020). However, pruning an initialized dense network before training often leads to a sparse network that is difficult to optimize and fails to match the original generalization (Li et al., 2017; Evci et al., 2022). This phenomenon led to a question, first posed by Frankle & Carbin (2019) as the lottery ticket hypothesis (LTH): Is there a sparse subnetwork (i.e., a winning lottery ticket) that can achieve generalization comparable to its dense counterpart when trained from initialization? This challenging research question has garnered significant attention and inspired many follow-up studies.

*Iterative magnitude pruning* (Frankle & Carbin, 2019) (IMP) is one of the representative methods to identify a winning ticket through iterating three phases: training, pruning, and rewinding. Many researchers have sought to understand how IMP finds a winning ticket. Among several insightful findings, perspectives from the loss landscape have provided valuable insights. Frankle et al. (2020a); Evci et al. (2022); Paul et al. (2023) have shown that IMP can find a winning ticket only when the network obtained after the training phase maintains its basin of attraction[1] after the pruning and

---

[1]Following previous works (Evci et al., 2022; Paul et al., 2023), we define a basin of attraction as a set of points connected by low-loss paths, where gradient descent from any point converges to the same minimum.

Figure 1: **Illustration of our motivation and method.** $\psi$ and $\phi$ denote network parameters of normalization layers and parameters excluding those of normalization layers, respectively. The 'LMC region' refers to a region of solutions that are linearly mode-connected to the LRR or AWS solution.

rewinding phases, thereby preserving strong generalization potential (i.e., generalization ability after training) throughout subsequent iterations. Since this condition is difficult to satisfy in relatively large-scale settings, variants of IMP bypass the challenges by either rewinding to warm-up trained parameters (*weight rewinding*) (Frankle et al., 2020a) or skipping the parameter rewinding phase (*learning rate rewinding*) (Renda et al., 2020). Although they found a subnetwork that performs comparably to a dense network after training, it is not at initialization, and thus not a winning ticket.

In this paper, we empirically show that an effective signed mask, a sparse mask with parameter sign information, is a key to satisfying the challenging condition for finding a winning ticket. Specifically, we leverage learning rate rewinding (LRR) for its ability to find effective parameter sign and sparsity configuration (Gadhikar & Burkholz, 2024). Then, with a slight modification to LRR, we demonstrate that if parameter signs are preserved, the subnetwork obtained through our LRR variant remains within its basin of attraction even after randomly initializing its parameters. This implies that the generalization potential of the subnetwork can be transferred to any randomly initialized network via the signed mask, possibly allowing it to generalize comparably to the dense network after training.

We observe that the original LRR fails to achieve this. As illustrated on the left side of Figure 1, we observe that the LRR subnetwork leaves its basin after randomly initializing its parameters while preserving their signs, indicated by the red ball and the high error barrier between the yellow and red balls, similar to the findings in Frankle et al. (2020b). This failure stems from the significant influence of initializing normalization layer parameters. Interestingly, when we exclude normalization parameters from initialization and only randomize the other parameters while preserving their signs, the resulting network retains the original basin. This, in turn, leads to convergence to a solution with a low error barrier along the linear path to the LRR solution (i.e., the solution obtained by training the LRR subnetwork), as illustrated by the green ball moving towards the blue ball on the orange line. These results suggest that the signed mask and normalization parameters of the LRR subnetwork enable any randomly initialized network to inherit its generalization potential.

To take a step closer to finding a winning ticket, we eliminate the need for trained normalization parameters by addressing the adverse effects of initializing them. To this end, we propose AWS, a variation of LRR to find **A W**inning **S**ign, which prevents high error barriers along the linear path between the AWS subnetwork and its counterpart with initialized normalization parameters. During training, AWS randomly interpolates between the current and initialized normalization parameters linearly, using the interpolated values for each forward pass. As illustrated on the right side of Figure 1, we argue that the AWS subnetwork stays within its basin even after randomizing parameters while preserving their signs, indicated by the low error barrier between the yellow and red balls. This helps the resulting network converge to a solution linearly mode-connected to the AWS solution, with performance close to the dense network (gray dotted line). Experiments across architectures and datasets show that the signed mask from the AWS subnetwork enables any randomly initialized network to generalize comparably to the dense counterpart after training.

We summarize the contributions of our work as follows:

- We observe that any randomly initialized network can inherit the generalization potential of the LRR subnetwork through its signed mask and the normalization layer parameters.
- We propose AWS that alleviates the dependence on trained normalization parameters by preventing high error barriers between the AWS subnetwork and its counterpart with initialized normalization parameters.
- In contrast to existing methods that are limited to finding a winning ticket with an ad-hoc initialization, we show that any randomly initialized network can generalize comparably to a dense network after training by applying the AWS-driven signed mask to it.

## 2 RELATED WORKS

**Lottery Ticket Hypothesis (LTH).** Frankle & Carbin (2019) propose LTH which states that within a dense network, there exists a sparse subnetwork that, when trained from initialization, can achieve performance comparable to the dense counterpart. To find such a winning lottery ticket, the authors proposed *iterative magnitude pruning* (IMP) and demonstrated that IMP successfully finds a winning ticket in a relatively small-scale setting. Follow-up works have delved into a broad range of topics related to LTH, such as theoretical support for the existence of the winning ticket (Malach et al., 2020; Orseau et al., 2020; Burkholz, 2022; da Cunha et al., 2022), efficient alternatives to IMP (You et al., 2020), searching for a winning ticket without weight training (Chen et al., 2022; Sreenivasan et al., 2022; Koster et al., 2022), and empirical analyses on winning ticket (Zhou et al., 2019; Frankle et al., 2020a; Ma et al., 2021; Sakamoto & Sato, 2022; Evci et al., 2022; Paul et al., 2023).

**Insights into a Winning Ticket.** One of the most important topics is investigating what makes a sparse network win the lottery. Frankle et al. (2020a) introduced the notion of mode connectivity (Freeman & Bruna, 2017; Nguyen, 2019; Draxler et al., 2018; Garipov et al., 2018; Lubana et al., 2023) into LTH to investigate the conditions under which IMP finds a winning ticket. They consider a network stable against SGD noise if, under different SGD randomness, it converges to a region of solutions that exhibit low error barriers along the linear path connecting them. Based on this definition, they demonstrated that IMP succeeds only when the rewound network is stable against SGD noise. Evci et al. (2022) found that a winning ticket can be found only when it resides in the same basin as the pruning solution used to obtain the sparse pruning mask. Paul et al. (2023) demonstrated that a sparse mask obtained in an IMP iteration guides the subsequent pruning solution to be linearly mode-connected to the previous IMP solution, leading the consecutive pruning solutions to be piece-wise linearly mode-connected. In summary, these findings suggest that IMP can identify a winning ticket only when the network obtained from the training phase remains within its basin of attraction after the pruning and rewinding phases, thereby preserving the generalization of the original dense network throughout all iterations. Variants of IMP, such as *weight rewinding* (Frankle et al., 2020a) and *learning rate rewinding* (Renda et al., 2020), bypass this challenging condition by applying the found sparse mask to trained parameters rather than initialized ones, failing to find a winning ticket.

**Significance of Parameter Signs in LTH.** Recently, several works reported the importance of parameter signs from the perspective of representation capacity (Wang et al., 2023a;b). Zhou et al. (2019) are the first to discover the role of parameter signs in the context of LTH. They empirically showed that in the parameter rewinding stage of IMP, rewinding parameter signs has a greater impact on the performance than the magnitudes. By contrast, Frankle et al. (2020b) showed that transferring the signed mask obtained by IMP to the original initialization performs worse than transferring them along with the respective magnitudes. Gadhikar & Burkholz (2024) demonstrated that IMP can fail to find a winning ticket because it loses crucial sign information during parameter rewinding and struggles to learn an effective sign configuration again due to reduced network capacity. They claimed that learning rate rewinding (Renda et al., 2020) (LRR), on the other hand, identifies more performant sparse networks by finding and maintaining the effective sign configuration during training.

We observe that the ineffectiveness of transferring parameter signs to an initialized network, as noted in Frankle et al. (2020b), is due to the adverse effect of initializing the normalization layer parameters. To address this issue, we propose a slight variant of LRR and demonstrate that the challenging conditions for finding a winning ticket suggested by Frankle et al. (2020a); Evci et al. (2022); Paul et al. (2023) can be satisfied using the effective signed mask acquired by our LRR variant.

## 3 METHOD

### 3.1 NOTATIONS AND BACKGROUND

**Lottery Ticket Hypothesis.** Let $\boldsymbol{\theta} \in \mathbb{R}^d$ denote the parameters of a neural network. We use $\boldsymbol{\theta}$ to also represent the neural network parameterized by $\boldsymbol{\theta}$. Consider a binary mask $\boldsymbol{m} \in \{0, 1\}^d$, having the same shape as $\boldsymbol{\theta}$. Note that the mask values for parameters not targeted for pruning, such as biases, are fixed to be 1. Then, the mask $\boldsymbol{m}$ defines a sparse subnetwork of the original dense network as $\boldsymbol{\theta} \odot \boldsymbol{m}$. The lottery ticket hypothesis (LTH) (Frankle & Carbin, 2019) posits the existence of a mask with non-trivial sparsity (i.e., $\sum_i m_i \ll d$) that allows $\boldsymbol{\theta}_{\text{init}} \odot \boldsymbol{m}$, where $\boldsymbol{\theta}_{\text{init}}$

represents initialized parameters, to perform comparably to the dense network after training. Such a sparse network at initialization is referred to as a *winning ticket*. *Iterative magnitude pruning* (IMP) (Frankle & Carbin, 2019) was suggested as a way to find the winning ticket through iterative `training → pruning → rewinding` procedures. Let $\mathcal{A}(\boldsymbol{\theta}, u)$ represent an SGD learning algorithm with an initialized learning rate scheduler that updates $\boldsymbol{\theta}$ until convergence using SGD randomness $u \sim U$ (e.g. randomness from a data loader or data augmentations). We omit $u$ from $\mathcal{A}(\boldsymbol{\theta}, u)$ if unnecessary. At `training` phase of the $t$-th iteration, IMP trains the masked initial parameters, obtaining $\boldsymbol{\theta}_t^{\text{IMP}} \odot \boldsymbol{m}_{t-1}^{\text{IMP}} = \mathcal{A}(\boldsymbol{\theta}_0^{\text{IMP}} \odot \boldsymbol{m}_{t-1}^{\text{IMP}})$, where $\boldsymbol{m}_{t-1}^{\text{IMP}}$ denotes the mask obtained from the $(t-1)$-th iteration. At `pruning` phase, IMP produces the $t$-th mask by removing a portion of the non-zero weights (commonly 20%) in $\boldsymbol{\theta}_t^{\text{IMP}} \odot \boldsymbol{m}_{t-1}^{\text{IMP}}$ with the smallest magnitudes: $\boldsymbol{m}_t^{\text{IMP}} = \texttt{prune}(\boldsymbol{\theta}_t^{\text{IMP}} \odot \boldsymbol{m}_{t-1}^{\text{IMP}})$. At `rewinding` phase, $\boldsymbol{\theta}_t^{\text{IMP}}$ is rewound to the initial parameters, $\boldsymbol{\theta}_0^{\text{IMP}}$, and the entire process is repeated until $t = T$. Finally, IMP produces $\boldsymbol{\theta}_0^{\text{IMP}} \odot \boldsymbol{m}_T^{\text{IMP}}$, referred to as the *IMP subnetwork*, and after training the IMP subnetwork, IMP obtains $\mathcal{A}(\boldsymbol{\theta}_0^{\text{IMP}} \odot \boldsymbol{m}_T^{\text{IMP}})$, referred to as the *IMP solution*.

**Variants of IMP.** Several variants of IMP have been proposed to address the failure of IMP in generalizing to more challenging settings, especially focusing on the `rewinding` phase. Based on the analysis of stability against SGD noise, Frankle et al. (2020a) proposed *weight rewinding* (WR) that rewinds $\boldsymbol{\theta}_t^{\text{IMP}}$ to the warm-up trained parameters rather than to $\boldsymbol{\theta}_0^{\text{IMP}}$. *Learning rate rewinding* (LRR) (Renda et al., 2020) is another variant of IMP that skips parameter rewinding and instead rewinds only the learning rate schedule. At the `training` phase of the $t$-th iteration, LRR trains $\boldsymbol{\theta}_{t-1}^{\text{LRR}} \odot \boldsymbol{m}_{t-1}^{\text{LRR}}$, obtaining $\boldsymbol{\theta}_t^{\text{LRR}} \odot \boldsymbol{m}_{t-1}^{\text{LRR}} = \mathcal{A}(\boldsymbol{\theta}_{t-1}^{\text{LRR}} \odot \boldsymbol{m}_{t-1}^{\text{LRR}})$, where $\boldsymbol{m}_{t-1}^{\text{LRR}}$ represents the mask obtained from the $(t-1)$-th iteration. After obtaining $\boldsymbol{m}_t^{\text{LRR}} = \texttt{prune}(\boldsymbol{\theta}_t^{\text{LRR}} \odot \boldsymbol{m}_{t-1}^{\text{LRR}})$ at the `pruning` phase, LRR skips the `rewinding` phase and continues the subsequent iterations until $t = T$. Finally, LRR produces $\boldsymbol{\theta}_T^{\text{LRR}} \odot \boldsymbol{m}_T^{\text{LRR}}$, referred to as the *LRR subnetwork*, and after training the LRR subnetwork, LRR obtains $\mathcal{A}(\boldsymbol{\theta}_T^{\text{LRR}} \odot \boldsymbol{m}_T^{\text{LRR}})$, referred to as the *LRR solution*.

**Linear Mode Connectivity and stability against SGD Noise.** Linear mode connectivity and stability against SGD noise have been adopted as useful tools for analyzing winning tickets from the loss landscape perspective (Frankle et al., 2020a; Evci et al., 2022; Paul et al., 2023). Let $\mathcal{E}(\boldsymbol{\theta})$ denote the test error of $\boldsymbol{\theta}$. We define the error barrier between two parameters, $\boldsymbol{\theta}$ and $\boldsymbol{\theta}'$, when interpolating them by a factor of $\alpha$ as the difference between the error of the interpolated network and the mean error:

$$\mathcal{E}_\alpha(\boldsymbol{\theta}, \boldsymbol{\theta}') = \mathcal{E}(\alpha\boldsymbol{\theta} + (1-\alpha)\boldsymbol{\theta}') - (\mathcal{E}(\boldsymbol{\theta}) + \mathcal{E}(\boldsymbol{\theta}'))/2. \tag{1}$$

Then, we define $\mathcal{E}(\boldsymbol{\theta}, \boldsymbol{\theta}') = \sup_\alpha \mathcal{E}_\alpha(\boldsymbol{\theta}, \boldsymbol{\theta}')$. If $\mathcal{E}(\boldsymbol{\theta}, \boldsymbol{\theta}')$ is smaller than a sufficiently small value $\epsilon$, we consider $\boldsymbol{\theta}$ and $\boldsymbol{\theta}'$ to be linearly mode-connected (LMC). $\epsilon$ is often determined empirically such as the standard deviation of errors of a dense network across different training seeds (Paul et al., 2023). Based on the definition of LMC, we define the stability against SGD noise as the condition where a pair of networks are LMC when trained from the same initial parameters but with different SGD randomness (Frankle et al., 2020a). Formally, $\boldsymbol{\theta}$ is considered stable against SGD noise if $\mathcal{A}(\boldsymbol{\theta}, u)$ and $\mathcal{A}(\boldsymbol{\theta}, u')$ are LMC with $u, u' \sim U$.

As demonstrated by previous works (Frankle et al., 2020a; Evci et al., 2022; Paul et al., 2023), IMP can identify a winning ticket only when at the $t$-th iteration, $\boldsymbol{\theta}_0^{\text{IMP}} \odot \boldsymbol{m}_{t-1}^{\text{IMP}}$ still resides in the basin of attraction of the $(t-1)$-th solution after the `training` phase, $\boldsymbol{\theta}_{t-1}^{\text{IMP}} \odot \boldsymbol{m}_{t-2}^{\text{IMP}}$, even after updating $\boldsymbol{m}_{t-2}^{\text{IMP}}$ to $\boldsymbol{m}_{t-1}^{\text{IMP}}$ and rewinding $\boldsymbol{\theta}_{t-1}^{\text{IMP}}$ to $\boldsymbol{\theta}_0^{\text{IMP}}$. Paul et al. (2023) claim that if this condition is satisfied at every iteration, the consecutive solutions after the training phase are piece-wise LMC, allowing the final IMP solution to generalize comparably to the dense network. To examine this condition, previous works investigate whether a rewound network is stable against SGD noise and converges to a solution with linear mode connectivity to the network before the rewinding and pruning phases. WR and LRR bypass the challenging condition by rewinding to warm-up trained parameters or skipping the `rewinding` phase. In contrast, this paper argues that any randomly initialized network can inherit strong generalization potential through a mask with parameter sign information, progressing toward the goal of LTH.

## 3.2 MOTIVATION

Gadhikar & Burkholz (2024) demonstrated that learning rate rewinding (LRR) successfully maintains the performance of dense networks by learning and maintaining an effective parameter sign

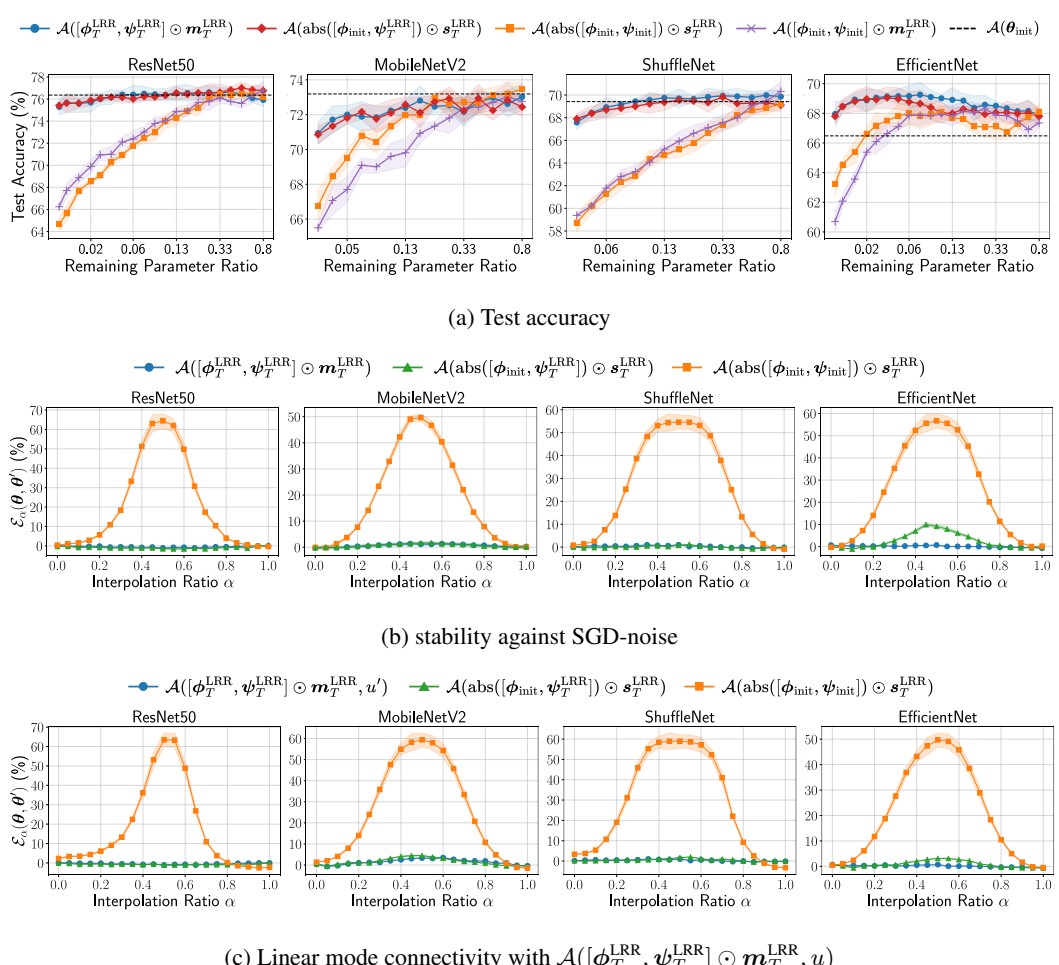

(a) Test accuracy

(b) stability against SGD-noise

(c) Linear mode connectivity with $\mathcal{A}([\boldsymbol{\phi}_T^{\mathrm{LRR}}, \boldsymbol{\psi}_T^{\mathrm{LRR}}] \odot \boldsymbol{m}_T^{\mathrm{LRR}}, u)$

Figure 2: **Motivational experiments on CIFAR-100.** We investigate the effect of parameter initialization in the LRR subnetwork while preserving their signs with respect to **(a) test accuracy**, **(b) SGD-noise stability**, and **(c) linear mode connectivity with the LRR subnetwork.** In (b) and (c), we use a pruned network with a remaining parameter ratio of approximately 0.09. We show the mean (each point) and standard deviation (shaded area) across 3 trials.

configuration while pruning unimportant parameters, which iterative magnitude pruning (IMP) and weight rewinding (WR) fail. Several other works (Zhou et al., 2019; Frankle et al., 2020b; Chen et al., 2022; Sreenivasan et al., 2022; Koster et al., 2022; Wang et al., 2023b;a) also found the importance of parameter signs in the context of LTH or representation learning. In this work, motivated by these studies, we hypothesize that **the parameter sign information obtained through LRR can transfer the generalization potential of the LRR subnetwork to any randomly initialized network.** Let $\mathrm{sign}_0(\cdot)$ denote a function that outputs the sign of each input element if it is non-zero, and 0 otherwise. Then, $\boldsymbol{s}_T^{\mathrm{LRR}} = \mathrm{sign}_0(\boldsymbol{\theta}_T^{\mathrm{LRR}} \odot \boldsymbol{m}_T^{\mathrm{LRR}})$ represents the signed mask of the LRR subnetwork. More specifically, our hypothesis states that applying $\boldsymbol{s}_T^{\mathrm{LRR}}$ to a randomly initialized network, $\boldsymbol{\theta}_{\mathrm{init}}$, will enable the resulting network to match the performance of the LRR solution after training: $\mathcal{A}(\mathrm{abs}(\boldsymbol{\theta}_{\mathrm{init}}) \odot \boldsymbol{s}_T^{\mathrm{LRR}}) \approx \mathcal{A}(\boldsymbol{\theta}_T^{\mathrm{LRR}} \odot \boldsymbol{m}_T^{\mathrm{LRR}})$ where $\mathrm{abs}(\cdot)$ denotes the modulus function.

A similar idea was studied by Frankle et al. (2020b). They showed that replacing the signs of initialized parameters with those of the IMP subnetwork does not improve performance compared to using magnitude information in conjunction. We point out that this failure is attributed to ignoring the impact of parameters that may rely more on magnitudes than their signs. In the case of weight parameters, such as the weights in a convolutional layer, the sign configuration has a critical role in determining the functional mechanism of the layer as discussed in Wang et al. (2023b;a); Gadhikar & Burkholz (2024). By contrast, for parameters in a normalization layer, such as the batch normalization (Ioffe & Szegedy, 2015) or layer normalization (Ba et al., 2016), the magnitude may be much more important

---

**Algorithm 1** AWS: a slight modification to LRR to find **w**inning **s**ign. The modification is highlighted in red.

---

**Require:** Initialize $\boldsymbol{\phi}_0$, $\boldsymbol{\psi}_0$, and $\boldsymbol{m}_0 \leftarrow (1, \ldots, 1) \in \mathbb{R}^d$

1: **for** $t = 1$ to $T$ **do**
2:    **while** not converge **do**
3:       $(\boldsymbol{\psi}_{t-1}, \boldsymbol{\psi}_{\text{init}})_\alpha = \alpha \cdot \boldsymbol{\psi}_{t-1} + (1 - \alpha) \cdot \boldsymbol{\psi}_{\text{init}}$ where $\alpha \sim U(0, 1)$    $\triangleright$ Interpolating $\boldsymbol{\psi}_{t-1}$ and $\boldsymbol{\psi}_{\text{init}}$
4:       Forward pass using $[\boldsymbol{\phi}_{t-1}, (\boldsymbol{\psi}_{t-1}, \boldsymbol{\psi}_{\text{init}})_\alpha] \odot \boldsymbol{m}_{t-1}$ $\triangleright$ Forward pass with the interpolated parameters
5:       Update $\boldsymbol{\phi}_{t-1}$ and $\boldsymbol{\psi}_{t-1}$ via gradient descent
6:    **end while**
7:    $\boldsymbol{\phi}_t \leftarrow \boldsymbol{\phi}_{t-1}$ and $\boldsymbol{\psi}_t \leftarrow \boldsymbol{\psi}_{t-1}$
8:    $\boldsymbol{m}_t \leftarrow \texttt{prune}([\boldsymbol{\phi}_t, \boldsymbol{\psi}_t] \odot \boldsymbol{m}_{t-1})$    $\triangleright$ Update the sparse mask
9:    Rewind learning rate scheduler
10: **end for**
11: **return** $\text{sign}_0([\boldsymbol{\phi}_T, \boldsymbol{\psi}_T] \odot \boldsymbol{m}_T)$    $\triangleright$ Obtain the signed mask

---

than the sign since the scaling parameter, initialized to 1, is nearly always positive after training, and the bias parameter loses the replaced signs since it is initialized to 0. We also observe that transferring the signs of parameters of normalization layers is not beneficial as analyzed in Appendix A.

Let $\phi$ and $\psi$ denote the parameters excluding those of normalization layers and those of normalization layers, respectively. Then, we represent a randomly initialized network and the LRR subnetwork as $\boldsymbol{\theta}_{\text{init}} = [\boldsymbol{\phi}_{\text{init}}, \boldsymbol{\psi}_{\text{init}}]$ and $\boldsymbol{\theta}_T^{\text{LRR}} = [\boldsymbol{\phi}_T^{\text{LRR}}, \boldsymbol{\psi}_T^{\text{LRR}}]$, respectively. To test our conjecture, we compare two cases: in the LRR subnetwork, randomly initializing the magnitudes of both $\boldsymbol{\phi}_T^{\text{LRR}}$ and $\boldsymbol{\psi}_T^{\text{LRR}}$ versus only $\boldsymbol{\phi}_T^{\text{LRR}}$ while maintaining the signed mask (i.e., $\mathcal{A}(\text{abs}([\boldsymbol{\phi}_{\text{init}}, \boldsymbol{\psi}_{\text{init}}]) \odot \boldsymbol{s}_T^{\text{LRR}})$ vs. $\mathcal{A}(\text{abs}([\boldsymbol{\phi}_{\text{init}}, \boldsymbol{\psi}_T^{\text{LRR}}]) \odot \boldsymbol{s}_T^{\text{LRR}})$). Figure 2 shows the results on CIFAR-100 with various architectures. For details on the experiments, please refer to Section 4.1. Figure 2a shows that similar to the results in Frankle et al. (2020b), preserving the signed mask while initializing the magnitudes of all parameters randomly (indicated by the orange plots) results in performance similar to the case where sign information is not used (indicated by the purple plots), lagging far behind the performance of the LRR solution (indicated by the blue plots). This result indicates that sign information is not beneficial when all parameters are randomly initialized. On the other hand, interestingly, when the parameters of normalization layers are kept intact (indicated by the red plots), the performance is comparable to the LRR solution after training, indicating that using the sign information of the LRR subnetwork is beneficial when excluding the influence of initializing the normalization layer parameters. To examine whether the resulting networks reside in the basin of attraction of the LRR subnetwork, we analyze their stability against SGD noise and linear mode connectivity with the LRR subnetwork. Figure 2b demonstrates that while randomly initializing the magnitudes of all parameters significantly ruins the stability of the LRR subnetwork (indicated by the orange plots), it is effectively preserved when ignoring the influence of initializing the normalization layer parameters (indicated by the green plots). As shown in Figure 2c, we also observe that the preserved stability, in turn, leads the resulting network, $\text{abs}([\boldsymbol{\phi}_{\text{init}}, \boldsymbol{\psi}_T^{\text{LRR}}]) \odot \boldsymbol{s}_T^{\text{LRR}}$, to converge to a solution with a low error barrier along the linear path connecting it to the LRR solution (indicated by the green plots), suggesting they are in the same basin of attraction. This contrasts with the high error barrier observed when initializing all parameter magnitudes (indicated by the orange plots).

The left side of Figure 1 illustrates the observations from our motivational experiments:

- In the LRR subnetwork, randomly initializing all parameters while preserving their signs causes the resulting network to lose SGD noise stability of the LRR subnetwork, potentially leading to a suboptimal solution, as indicated by the red ball.

- On the other hand, when the normalization layer parameters are kept intact, the resulting network, $\text{abs}([\boldsymbol{\phi}_{\text{init}}, \boldsymbol{\psi}_T^{\text{LRR}}]) \odot \boldsymbol{s}_T^{\text{LRR}}$, exhibits significant stability against SGD noise and converges to a solution with a low error barrier along the linear path connecting it to the LRR solution, resulting in performance comparable to that of dense network, as indicated by the green ball.

Our observations suggest that when the parameter signs are preserved, the LRR subnetwork stays within its basin of attraction even after randomly initializing the parameters, excluding those of normalization layers. In other words, a randomly initialized network can inherit the generalization potential of the LRR subnetwork through the signed mask and normalization layer parameters of the LRR subnetwork.

### 3.3 AWS: Finding A Winning Sign

Our observations provide valuable insights into the role of the signed mask in transferring strong generalization potential to any randomly initialized network. However, the need for the trained normalization parameters still limits LRR in achieving the goal of LTH. In this subsection, we introduce a method that addresses the adverse impact of initializing the normalization layer parameters to further progress toward the goal of LTH. Our goal is to maintain the basin of attraction in which the LRR subnetwork resides when the normalization layer parameters are initialized. Two networks residing in the same basin may indicate that no high error barrier exists along the linear path connecting them (Evci et al., 2022). Thus, we propose a simple variation of LRR to find **a w**inning **s**ign, referred to as **AWS**, that prevents any high error barriers along the linear path connecting the LRR subnetwork and its counterpart with initialized normalization parameters. Specifically, AWS randomly and linearly interpolates the parameters of normalization layers with their initialization and uses the interpolated parameters instead of the original parameters during training. Formally, at every network forward pass during the $t$-th iteration, AWS obtains

$$(\boldsymbol{\psi}_t^{\mathrm{AWS}}, \boldsymbol{\psi}_{\mathrm{init}})_\alpha = \alpha \cdot \boldsymbol{\psi}_t^{\mathrm{AWS}} + (1 - \alpha) \cdot \boldsymbol{\psi}_{\mathrm{init}}, \tag{2}$$

where $\boldsymbol{\psi}_t^{\mathrm{AWS}}$ denotes the parameters of normalization layers during the $t$-th iteration of AWS and $\alpha \sim U(0, 1)$. Then, AWS uses $(\boldsymbol{\psi}_t^{\mathrm{AWS}}, \boldsymbol{\psi}_{\mathrm{init}})_\alpha$ instead of $\boldsymbol{\psi}_t^{\mathrm{AWS}}$ for network forwarding. We present the pseudo-code of AWS in Algorithm 1, omitting the superscript 'AWS' for simplicity. After all iterations, we transfer the resulting signed mask, $\boldsymbol{s}_T^{\mathrm{AWS}}$, to a randomly initialized network, obtaining $\mathrm{abs}(\boldsymbol{\theta}_{\mathrm{init}}) \odot \boldsymbol{s}_T^{\mathrm{AWS}}$, and train it using normal training set-up until convergence.

The right side of Figure 1 illustrates the effectiveness of AWS. In contrast to LRR, AWS can allow a randomly initialized network to lie in the basin of attraction of the AWS subnetwork through the learned signed mask, $\boldsymbol{s}_T^{\mathrm{AWS}}$, possibly leading it to converge to a solution that is linearly mode-connected to the AWS solution (indicated by the red and blue ball). Thus, we argue that $\boldsymbol{s}_T^{\mathrm{AWS}}$ **can transfer the generalization potential of the AWS subnetwork to any randomly initialized network, possibly resulting in performance comparable to a dense network after training.**

## 4 Experiments

### 4.1 Implementation Details

**Datasets and models.** Following the previous works (Ma et al., 2021; Gadhikar & Burkholz, 2024), we conduct experiments on CIFAR-100 (Krizhevsky & Hinton, 2009), Tiny-ImageNet (Le & Yang, 2015), and ImageNet (Russakovsky et al., 2015). For both CIFAR-100 and Tiny-ImageNet, we adopt ResNet-50 (He et al., 2016), MobileNetV2 (Sandler et al., 2018), ShuffleNet (Zhang et al., 2018), and EfficientNet (Tan & Le, 2019) to validate our method across various architectures. For ImageNet experiments, we use ResNet50, MobileNetV2, and MLP-Mixer (Tolstikhin et al., 2021). We adopt MLP-Mixer, which includes layer normalization, to demonstrate the generalization of our method to different types of normalization layers rather than the batch normalization layer.
**Implementation.** In both learning rate rewinding (LRR) and the proposed AWS method, we observe that many training epochs and learning rate scheduling are unnecessary during the `training` phase, as the network converges quickly due to the absence of parameter rewinding, and learning rate scheduling has little impact on the performance. Thus, during the `training` phase for both LRR and AWS, we train a network for 10 epochs for CIFAR-100 and Tiny-ImageNet experiments, and 5 epochs for ImageNet experiments without learning rate scheduling. To ensure a network can converge at the early iterations, we perform warm-up training before the first iteration: 10 epochs for CIFAR-100 and Tiny-ImageNet, and 20 epochs for ImageNet. After the $T$-th iteration, we conduct final training of 100 epochs for CIFAR-100 and Tiny-ImageNet, 120 epochs for ResNet-50 and MobileNetV2 on ImageNet, and 200 epochs for MLP-Mixer.
**Optimization.** For the CIFAR-100 and Tiny-ImageNet experiments, we use the SGD optimizer with a momentum of 0.9, a weight decay of 5e-4, and an initial learning rate of 0.1, except for MobileNetV2, which uses a learning rate of 0.05. The learning rate decays by a factor of 0.1 at the 50th and 75th epochs over 100 epochs. For ImageNet experiments, we use the Adam optimizer with $\beta_1 = 0.9$, $\beta_2 = 0.999$, and an initial learning rate of 0.001 for both ResNet-50 and MLP-Mixer, and the SGD optimizer with a momentum of 0.9 and an initial learning rate of 0.05 for MobileNetV2. The weight decay is set to 5e-5 for MobileNetV2 and MLP-Mixer, and 1e-5 for ResNet-50. We use cosine annealing for learning rate scheduling. The batch size is set to 128 for CIFAR-100 and Tiny-ImageNet, 256 for ResNet-50 and MobileNetV2 on ImageNet, and 512 for MLP-Mixer.

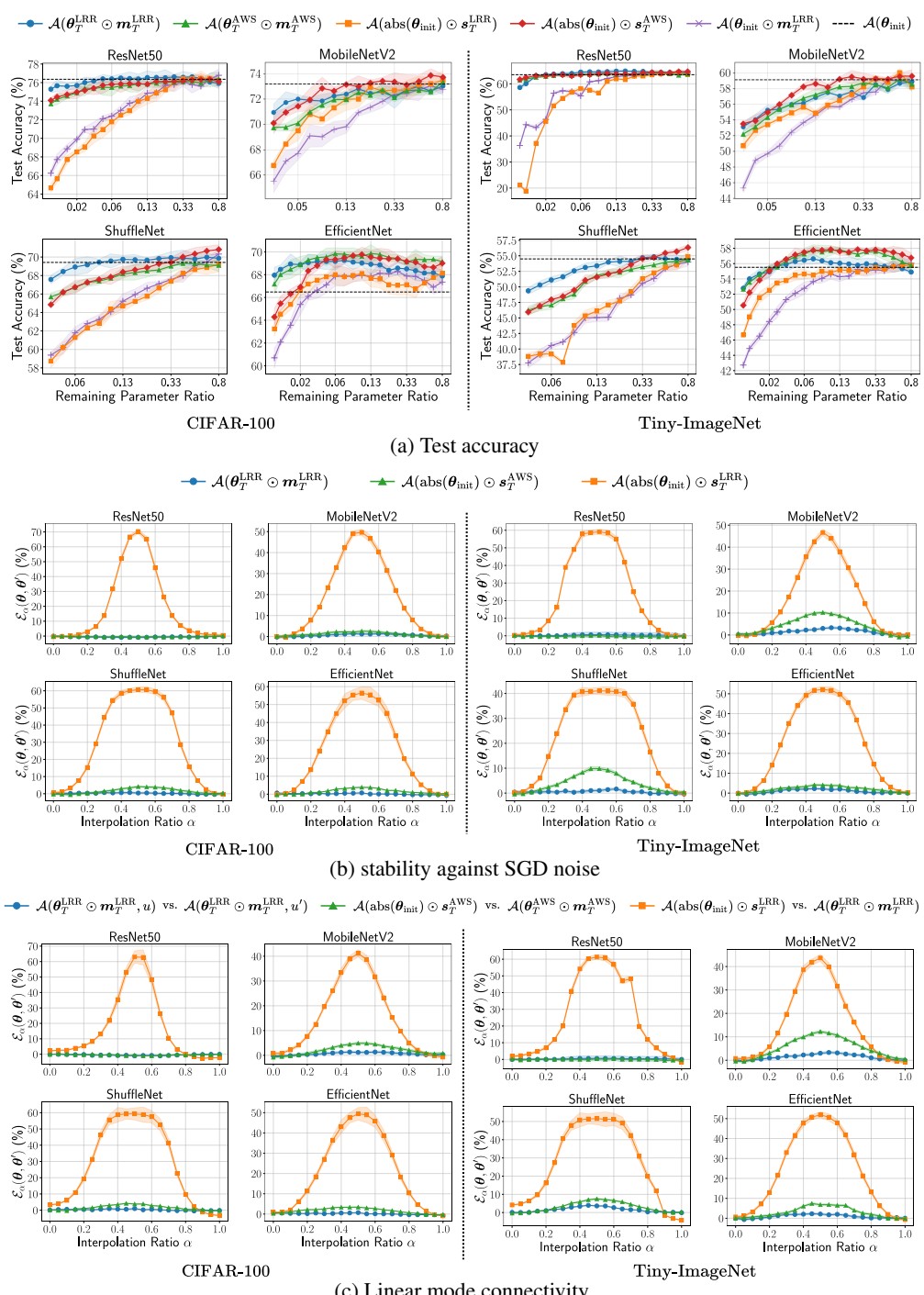

Figure 3: **Main results on CIFAR-100 and Tiny-ImageNet. (a):** Test accuracy of the LRR solution (blue), the AWS solution (green), a randomly initialized network trained with the LRR-driven signed mask (orange), and a randomly initialized network trained with the AWS-driven signed mask (red). **(b) and (c):** Analysis of SGD noise stability and linear mode connectivity, respectively. A randomly initialized network trained with the AWS-driven signed mask exhibits high SGD noise stability and low error barriers along the linear path to the AWS solution (green), contrasting to the case of LRR (orange). In (b) and (c), we use a pruned network with a remaining parameter ratio of approximately 0.09. We report the mean (each point) and standard deviation (shaded area) across 3 trials.

## 4.2 RESULTS ON CIFAR-100 AND TINY-IMAGENET

**Test Performance.** In Figure 3a, we report the test performance on CIFAR-100 and Tiny-ImageNet. We compare the performance of six networks after training: (1) initialized dense network, $\mathcal{A}(\boldsymbol{\theta}_{\text{init}})$, (2) LRR subnetwork, $\mathcal{A}(\boldsymbol{\theta}_T^{\text{LRR}} \odot \boldsymbol{m}_T^{\text{LRR}})$, (3) AWS subnetwork, $\mathcal{A}(\boldsymbol{\theta}_T^{\text{AWS}} \odot \boldsymbol{m}_T^{\text{AWS}})$, (4) randomly initialized network masked with a signed mask of a LRR subnetwork, $\mathcal{A}(\text{abs}(\boldsymbol{\theta}_{\text{init}}) \odot \boldsymbol{s}_T^{\text{LRR}})$, (5) randomly initialized network with a signed mask of a AWS subnetwork, $\mathcal{A}(\text{abs}(\boldsymbol{\theta}_{\text{init}}) \odot \boldsymbol{s}_T^{\text{AWS}})$, and (6) randomly initialized network masked with a mask of a LRR subnetwork, $\mathcal{A}(\boldsymbol{\theta}_{\text{init}} \odot \boldsymbol{m}_T^{\text{LRR}})$. Note that $\mathcal{A}(\cdot)$ indicates a normal training algorithm without interpolating normalization layer parameters. First, we note that in most cases, the AWS solution (indicated by the green plots) achieves performance comparable to or better than the LRR solution (indicated by the blue plots). This addresses the potential concern that randomly interpolating normalization parameters in our method could adversely affect the performance of the AWS solution. Then, we transfer the signed mask from LRR, $\boldsymbol{s}_T^{\text{LRR}}$, and AWS, $\boldsymbol{s}_T^{\text{AWS}}$, to a randomly initialized network $\boldsymbol{\theta}_{\text{init}}$. In the case of LRR, the performance of a randomly initialized network masked with $\boldsymbol{s}_T^{\text{LRR}}$ after training (indicated by the orange plots) is similar to the case when the sign information is not used (indicated by the purple plots), lagging far behind the performance of the LRR solution. On the other hand, we observe that in most cases, the performance of a randomly initialized network masked with $\boldsymbol{s}_T^{\text{AWS}}$ after training (indicated by the red plots) is comparable to that of the AWS solution. Finally, We observe that the signed mask obtained through AWS allows a randomly initialized network to perform comparably to a dense network (indicated by the dotted line) after training at non-trivial sparsity as long as the AWS solution performs comparably to the dense network.

**stability against SGD-Noise and Linear Mode Connectivity.** We further compare the effectiveness of $\boldsymbol{s}_T^{\text{AWS}}$ and $\boldsymbol{s}_T^{\text{LRR}}$ by examining whether they can transfer the strong generalization potential of the AWS or LRR subnetwork to a randomly initialized network. To this end, we first examine the SGD noise stability of $\text{abs}(\boldsymbol{\theta}_{\text{init}}) \odot \boldsymbol{s}_T^{\text{AWS}}$ and $\text{abs}(\boldsymbol{\theta}_{\text{init}}) \odot \boldsymbol{s}_T^{\text{LRR}}$. The results in Figure 3b demonstrate that $\text{abs}(\boldsymbol{\theta}_{\text{init}}) \odot \boldsymbol{s}_T^{\text{LRR}}$, shown by the orange plots, exhibit significantly high error barriers between a pair of networks trained with different SGD randomness. On the other hand, $\text{abs}(\boldsymbol{\theta}_{\text{init}}) \odot \boldsymbol{s}_T^{\text{AWS}}$, represented by the green plots, exhibits comparably low error barriers between a pair of networks trained with different SGD randomness, relative to the reference stability of $\boldsymbol{\theta}_T^{\text{LRR}} \odot \boldsymbol{s}_T^{\text{LRR}}$ shown by the blue plots. Moreover, we examine the linear mode connectivity between $\mathcal{A}(\text{abs}(\boldsymbol{\theta}_{\text{init}}) \odot \boldsymbol{s}_T^{\text{AWS}})$ and $\mathcal{A}(\text{abs}(\boldsymbol{\theta}_T^{\text{AWS}}) \odot \boldsymbol{s}_T^{\text{AWS}})$, as well as between $\mathcal{A}(\text{abs}(\boldsymbol{\theta}_{\text{init}}) \odot \boldsymbol{s}_T^{\text{LRR}})$ and $\mathcal{A}(\text{abs}(\boldsymbol{\theta}_T^{\text{LRR}}) \odot \boldsymbol{s}_T^{\text{LRR}})$ to confirm whether they lie in the same basin. The results in Figure 3c demonstrate that transferring $\boldsymbol{s}_T^{\text{LRR}}$ to a randomly initialized network causes the network to converge to a solution with significantly high error barriers between the LRR solution, as indicated by the orange plots. By contrast, a random initialized network masked with $\boldsymbol{s}_T^{\text{AWS}}$ exhibits relatively low error barriers between the AWS solution, as indicated by the green plots, compared to the reference shown by the blue plots.

**Discussion.** Our motivational experiments in Figure 2 show that the signed mask of the LRR subnetwork alone cannot transfer the generalization potential of the LRR subnetwork to a randomly initialized network; it is also necessary to transfer the normalization layer parameters of the LRR subnetwork. For the goal of LTH, we propose AWS that eliminates the need for the trained normalization parameters. The test performance in Figure 3a and the analysis of the SGD noise stability and linear mode connectivity in Figure 3b and Figure 3c show that a randomly initialized network masked with the signed mask from the AWS subnetwork lies in the basin of attraction of the AWS subnetwork. This demonstrates that $\boldsymbol{s}_T^{\text{AWS}}$ itself can transfer the generalization potential of the AWS subnetwork.

## 4.3 GENERALIZATION TO IMAGENET AND LAYER NORMALIZATION

To validate the effectiveness of AWS on a more challenging task, we evaluate AWS on ImageNet dataset. The results in Table 1 show that across various model and sparsity (i.e., 1- 'Remaining Params'), a randomly initialized network masked with the AWS-driven signed mask achieves performance comparable to the AWS solution (i.e., $\mathcal{A}(\text{abs}(\boldsymbol{\theta}_{\text{init}}) \odot \boldsymbol{s}_T^{\text{AWS}})$ vs. $\mathcal{A}(\boldsymbol{\theta}_T^{\text{AWS}} \odot \boldsymbol{m}_T^{\text{AWS}})$), whereas masking with the LRR-driven signed mask fails to reach the performance of the LRR solution (i.e., $\mathcal{A}(\text{abs}(\boldsymbol{\theta}_{\text{init}}) \odot \boldsymbol{s}_T^{\text{LRR}})$ vs. $\mathcal{A}(\boldsymbol{\theta}_T^{\text{LRR}} \odot \boldsymbol{m}_T^{\text{LRR}})$). Notably, we observe a similar trend in the results on MLP-Mixer, which uses only layer normalization for normalization. These results show that the signed mask of the AWS subnetwork can transfer the generalization potential of the AWS subnetwork to a randomly initialized network on a more challenging dataset and with layer normalization, further demonstrating the effectiveness of AWS.

Table 1: **Experimental results on ImageNet.** $\theta_{\text{init}}$ indicates randomly initialized parameters and 'Remaining Params.' refers to the remaining parameter ratio. We present more results in Table 2.

| Model | Method | Use AWS subnetwork | Trained from $\theta_{\text{init}}$ | Top-1 Acc. (%) | Remaining Params. |
|---|---|---|---|---|---|
| | Dense Network | - | - | $75.26 \pm 0.27$ | 1 |
| | $\mathcal{A}(\theta_T^{\text{LRR}} \odot m_T^{\text{LRR}})$ | ✗ | ✗ | $75.11 \pm 0.27$ | |
| | $\mathcal{A}(\theta_T^{\text{AWS}} \odot m_T^{\text{AWS}})$ | ✔ | ✗ | $74.48 \pm 0.16$ | 0.27 |
| | $\mathcal{A}(\text{abs}(\theta_{\text{init}}) \odot s_T^{\text{LRR}})$ | ✗ | ✔ | $73.20 \pm 0.21$ | |
| ResNet50 | $\mathcal{A}(\text{abs}(\theta_{\text{init}}) \odot s_T^{\text{AWS}})$ | ✔ | ✔ | $\mathbf{74.65 \pm 0.20}$ | |
| | $\mathcal{A}(\theta_T^{\text{LRR}} \odot m_T^{\text{LRR}})$ | ✗ | ✗ | $74.58 \pm 0.25$ | |
| | $\mathcal{A}(\theta_T^{\text{AWS}} \odot m_T^{\text{AWS}})$ | ✔ | ✗ | $73.63 \pm 0.21$ | 0.14 |
| | $\mathcal{A}(\text{abs}(\theta_{\text{init}}) \odot s_T^{\text{LRR}})$ | ✗ | ✔ | $71.94 \pm 0.22$ | |
| | $\mathcal{A}(\text{abs}(\theta_{\text{init}}) \odot s_T^{\text{AWS}})$ | ✔ | ✔ | $\mathbf{73.65 \pm 0.12}$ | |
| | Dense Network | - | - | $69.42 \pm 0.41$ | 1 |
| | $\mathcal{A}(\theta_T^{\text{LRR}} \odot m_T^{\text{LRR}})$ | ✗ | ✗ | $69.25 \pm 0.33$ | |
| | $\mathcal{A}(\theta_T^{\text{AWS}} \odot m_T^{\text{AWS}})$ | ✔ | ✗ | $68.75 \pm 0.22$ | 0.41 |
| | $\mathcal{A}(\text{abs}(\theta_{\text{init}}) \odot s_T^{\text{LRR}})$ | ✗ | ✔ | $67.99 \pm 0.11$ | |
| MobileNetV2 | $\mathcal{A}(\text{abs}(\theta_{\text{init}}) \odot s_T^{\text{AWS}})$ | ✔ | ✔ | $\mathbf{68.66 \pm 0.24}$ | |
| | $\mathcal{A}(\theta_T^{\text{LRR}} \odot m_T^{\text{LRR}})$ | ✗ | ✗ | $67.76 \pm 0.17$ | |
| | $\mathcal{A}(\theta_T^{\text{AWS}} \odot m_T^{\text{AWS}})$ | ✔ | ✗ | $66.17 \pm 0.20$ | 0.21 |
| | $\mathcal{A}(\text{abs}(\theta_{\text{init}}) \odot s_T^{\text{LRR}})$ | ✗ | ✔ | $64.03 \pm 0.21$ | |
| | $\mathcal{A}(\text{abs}(\theta_{\text{init}}) \odot s_T^{\text{AWS}})$ | ✔ | ✔ | $\mathbf{66.02 \pm 0.29}$ | |
| | Dense Network | - | - | $59.57 \pm 0.12$ | 1 |
| | $\mathcal{A}(\theta_T^{\text{LRR}} \odot m_T^{\text{LRR}})$ | ✗ | ✗ | $59.18 \pm 0.26$ | |
| | $\mathcal{A}(\theta_T^{\text{AWS}} \odot m_T^{\text{AWS}})$ | ✔ | ✗ | $59.23 \pm 0.18$ | 0.21 |
| | $\mathcal{A}(\text{abs}(\theta_{\text{init}}) \odot s_T^{\text{LRR}})$ | ✗ | ✔ | $57.54 \pm 0.30$ | |
| MLP-Mixer | $\mathcal{A}(\text{abs}(\theta_{\text{init}}) \odot s_T^{\text{AWS}})$ | ✔ | ✔ | $\mathbf{59.17 \pm 0.19}$ | |
| | $\mathcal{A}(\theta_T^{\text{LRR}} \odot m_T^{\text{LRR}})$ | ✗ | ✗ | $58.61 \pm 0.23$ | |
| | $\mathcal{A}(\theta_T^{\text{AWS}} \odot m_T^{\text{AWS}})$ | ✔ | ✗ | $58.77 \pm 0.11$ | 0.11 |
| | $\mathcal{A}(\text{abs}(\theta_{\text{init}}) \odot s_T^{\text{LRR}})$ | ✗ | ✔ | $56.91 \pm 0.21$ | |
| | $\mathcal{A}(\text{abs}(\theta_{\text{init}}) \odot s_T^{\text{AWS}})$ | ✔ | ✔ | $\mathbf{58.62 \pm 0.35}$ | |

## 4.4 COMPARISON TO RELATED WORK

There are existing works that also aim to search for an effective signed mask (Koster et al., 2022; Sreenivasan et al., 2022). Among them, we compare our AWS to GM (Sreenivasan et al., 2022), as GM is evaluated on a more diverse and large-scale architecture than the method of Koster et al. (2022). Figure 4 demonstrates that while GM performance falls short of that of the dense network across all levels of sparsity, a randomly initialized network masked with $s_T^{\text{AWS}}$ can perform comparably to both the AWS subnetwork and the dense network until a sparsity of about 0.9. We also highlight that our work provides valuable insights into the role of parameter signs and the influence of normalization layers concerning finding a winning ticket. Moreover, it is noteworthy that the signed mask obtained by AWS can be applied to any random initialization. In contrast, GM trains a signed mask tailored for a specific initialization, which likely limits its applicability to other initialization.

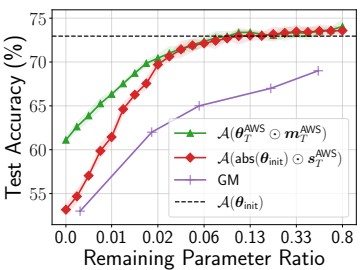

Figure 4: **Comparison to GM on CIFAR-100 with ResNet-32.** The results of GM are approximated from (Sreenivasan et al., 2022).

## 5 CONCLUSION

Finding a winning ticket is still an open problem in the field of the lottery ticket hypothesis (LTH). In this work, we show that an effective signed mask, a sparse mask with sign information, is crucial for conveying information that enables an initialized network to achieve strong generalization. We observe that the signed mask and normalization parameters from the subnetwork trained by learning rate rewinding (LRR) can transfer the generalization potential of the LRR subnetwork to any randomly initialized network. To progress towards the goal of LTH, we propose AWS, a slight variation of LRR, that mitigates the reliance on the trained normalization layer parameters by encouraging low error barriers between the AWS subnetwork and its counterpart with initialized normalization parameters. In contrast to the existing methods limited to finding a winning ticket with an ad-hoc initialization, we demonstrate that the signed mask from the AWS subnetwork can allow any randomly initialized network to reside within the basin of the AWS subnetwork, possibly leading the resulting network to generalize as well as the dense network. For future work, we will investigate the effectiveness of the signed mask acquired through AWS in the transfer learning scenario.

ACKNOWLEDGMENTS

This work was supported in part by the IITP grants [No.2021-0-01343, Artificial Intelligence Graduate School Program (Seoul National University), No. 2021-0-02068, and No.2023-0-00156], and the NOTIE grant (No. RS-2024-00432410) by the Korean government.

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

## A  IMPORTANCE OF PARAMETER SIGNS IN NORMALIZATION LAYERS.

In Section 3.2, we observe that preserving the signs of parameters in the LRR subnetwork while initializing their magnitudes, excluding those in normalization layers, allows the resulting network to stay within the basin of attraction of the LRR subnetwork, but it fails when the normalization layer parameters are initialized together. We claim that this occurs since the parameters in normalization layers rely more on the magnitude of parameters rather than their signs. The scaling factor in a normalization layer is nearly always positive, thus its sign information may be useless. In the case of the bias factor, it loses its sign information after initialized to 0, but it does not necessarily mean its sign information is not beneficial. To further validate our claim, we also compare the case where the bias factor in normalization layers is initialized to a constant, thus their sign information does not disappear after initialization. In Figure 5, we observe that maintaining the signs of bias factors (indicated by the purple plots) is not beneficial compared to the original initialization case (indicated by the orange plots), demonstrating our claim.

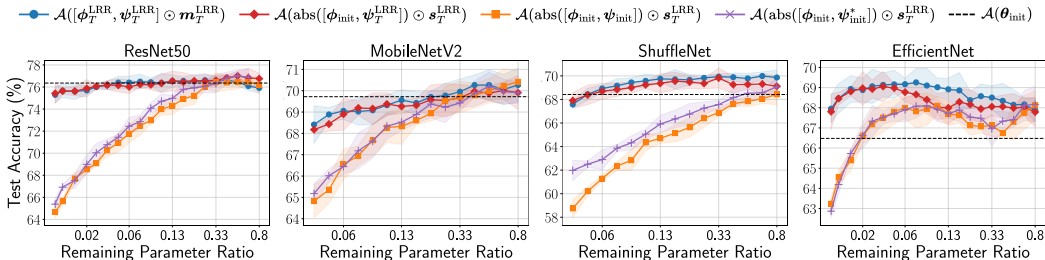

Figure 5: **Effect of transferring the sign of normalization layer parameters.** $\psi_{\text{init}}^*$ denotes the initialized normalization layer parameters whose scaling and bias factors are set to 1 and 0.1. We conduct the experiments on CIFAR-100.

## B  IS AWS ALWAYS BETTER THAN LRR?

In the main manuscript, we show that after training, the post-training performance of a randomly initialized network masked with $s_T^{\text{LRR}}$ (i.e., $\mathcal{A}(\text{abs}(\theta_{\text{init}}) \odot s_T^{\text{LRR}})$) lags far behind that of the LRR solution. However, we found that for several network architectures, the performance of $\mathcal{A}(\text{abs}(\theta_{\text{init}}) \odot s_T^{\text{LRR}})$ is similar to that of the LRR solution. In Figure 6a, we present the test performance on VGG11-bn (Simonyan & Zisserman, 2015). We observe that $\mathcal{A}(\text{abs}(\theta_{\text{init}}) \odot s_T^{\text{LRR}})$ (indicated by the orange plots) achieves performance similar to that of the LRR solution (indicated by the blue plots). Thus, the performance difference between $\mathcal{A}(\text{abs}(\theta_{\text{init}}) \odot s_T^{\text{LRR}})$ and $\mathcal{A}(\text{abs}(\theta_{\text{init}}) \odot s_T^{\text{AWS}})$ (indicated by the red plots) is trivial. We also investigate the SGD noise stability of $\mathcal{A}(\text{abs}(\theta_{\text{init}}) \odot s_T^{\text{LRR}})$ and $\mathcal{A}(\text{abs}(\theta_{\text{init}}) \odot s_T^{\text{AWS}})$ and their linear mode connectivity to the corresponding LRR or AWS solution in Figure 6b and Figure 6c, respectively. We observe that the signed masks from both LRR and AWS enable a randomly initialized network to remain stable against SGD noise and converge to a solution with linear mode connectivity to the corresponding LRR or AWS solution. Thus, in some cases, LRR can yield an effective signed mask that transfers the generalization potential of the LRR subnetwork to a randomly initialized network. However, we argue that AWS is more effective and generalizable to a wider range of more complex architectures as demonstrated in Figure 3.

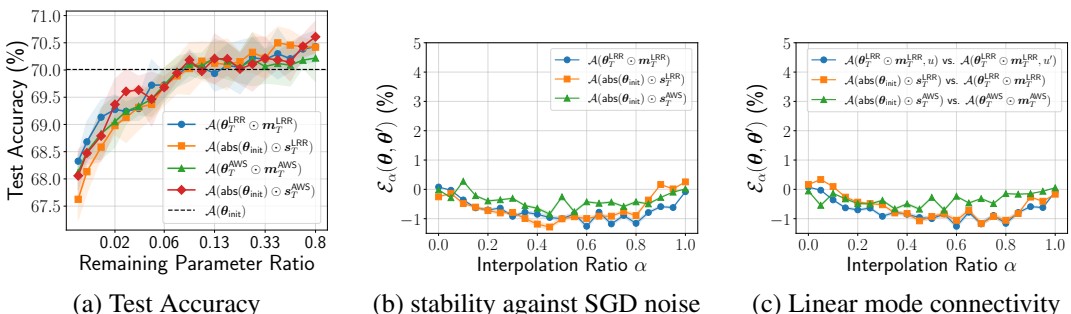

(a) Test Accuracy      (b) stability against SGD noise      (c) Linear mode connectivity

Figure 6: **Results of LRR on CIFAR-100 with VGG11-bn.**

Table 2: **Experimental results on ImageNet.** $\theta_{\text{init}}$ indicates randomly initialized parameters and 'Remaining Params.' refers to the remaining parameter ratio.

| Model | Method | Use AWS subnetwork | Trained from $\theta_{\text{init}}$ | Top-1 Acc. (%) | Remaining Params. |
|---|---|---|---|---|---|
| | Dense Network | - | - | $75.26 \pm 0.27$ | 1 |
| | $\mathcal{A}(\theta_T^{\text{LRR}} \odot m_T^{\text{LRR}})$ | ✗ | ✗ | $75.11 \pm 0.27$ | |
| | $\mathcal{A}(\theta_T^{\text{AWS}} \odot m_T^{\text{AWS}})$ | ✔ | ✗ | $74.48 \pm 0.16$ | 0.27 |
| | $\mathcal{A}(\text{abs}(\theta_{\text{init}}) \odot s_T^{\text{LRR}})$ | ✗ | ✔ | $73.20 \pm 0.21$ | |
| | $\mathcal{A}(\text{abs}(\theta_{\text{init}}) \odot s_T^{\text{AWS}})$ | ✔ | ✔ | $\mathbf{74.65 \pm 0.20}$ | |
| | $\mathcal{A}(\theta_T^{\text{LRR}} \odot m_T^{\text{LRR}})$ | ✗ | ✗ | $74.92 \pm 0.10$ | |
| | $\mathcal{A}(\theta_T^{\text{AWS}} \odot m_T^{\text{AWS}})$ | ✔ | ✗ | $74.06 \pm 0.18$ | 0.21 |
| | $\mathcal{A}(\text{abs}(\theta_{\text{init}}) \odot s_T^{\text{LRR}})$ | ✗ | ✔ | $72.96 \pm 0.20$ | |
| | $\mathcal{A}(\text{abs}(\theta_{\text{init}}) \odot s_T^{\text{AWS}})$ | ✔ | ✔ | $\mathbf{74.28 \pm 0.19}$ | |
| ResNet50 | $\mathcal{A}(\theta_T^{\text{LRR}} \odot m_T^{\text{LRR}})$ | ✗ | ✗ | $74.62 \pm 0.12$ | |
| | $\mathcal{A}(\theta_T^{\text{AWS}} \odot m_T^{\text{AWS}})$ | ✔ | ✗ | $73.76 \pm 0.26$ | 0.17 |
| | $\mathcal{A}(\text{abs}(\theta_{\text{init}}) \odot s_T^{\text{LRR}})$ | ✗ | ✔ | $72.46 \pm 0.18$ | |
| | $\mathcal{A}(\text{abs}(\theta_{\text{init}}) \odot s_T^{\text{AWS}})$ | ✔ | ✔ | $\mathbf{73.99 \pm 0.23}$ | |
| | $\mathcal{A}(\theta_T^{\text{LRR}} \odot m_T^{\text{LRR}})$ | ✗ | ✗ | $74.58 \pm 0.25$ | |
| | $\mathcal{A}(\theta_T^{\text{AWS}} \odot m_T^{\text{AWS}})$ | ✔ | ✗ | $73.63 \pm 0.21$ | 0.14 |
| | $\mathcal{A}(\text{abs}(\theta_{\text{init}}) \odot s_T^{\text{LRR}})$ | ✗ | ✔ | $71.94 \pm 0.22$ | |
| | $\mathcal{A}(\text{abs}(\theta_{\text{init}}) \odot s_T^{\text{AWS}})$ | ✔ | ✔ | $\mathbf{73.65 \pm 0.12}$ | |
| | Dense Network | - | - | $69.42 \pm 0.41$ | 1 |
| | $\mathcal{A}(\theta_T^{\text{LRR}} \odot m_T^{\text{LRR}})$ | ✗ | ✗ | $69.25 \pm 0.33$ | |
| | $\mathcal{A}(\theta_T^{\text{AWS}} \odot m_T^{\text{AWS}})$ | ✔ | ✗ | $68.75 \pm 0.22$ | 0.41 |
| | $\mathcal{A}(\text{abs}(\theta_{\text{init}}) \odot s_T^{\text{LRR}})$ | ✗ | ✔ | $67.99 \pm 0.11$ | |
| | $\mathcal{A}(\text{abs}(\theta_{\text{init}}) \odot s_T^{\text{AWS}})$ | ✔ | ✔ | $\mathbf{68.66 \pm 0.24}$ | |
| | $\mathcal{A}(\theta_T^{\text{LRR}} \odot m_T^{\text{LRR}})$ | ✗ | ✗ | $68.88 \pm 0.18$ | |
| | $\mathcal{A}(\theta_T^{\text{AWS}} \odot m_T^{\text{AWS}})$ | ✔ | ✗ | $68.13 \pm 0.32$ | 0.33 |
| | $\mathcal{A}(\text{abs}(\theta_{\text{init}}) \odot s_T^{\text{LRR}})$ | ✗ | ✔ | $67.01 \pm 0.19$ | |
| MobileNetV2 | $\mathcal{A}(\text{abs}(\theta_{\text{init}}) \odot s_T^{\text{AWS}})$ | ✔ | ✔ | $\mathbf{68.04 \pm 0.28}$ | |
| | $\mathcal{A}(\theta_T^{\text{LRR}} \odot m_T^{\text{LRR}})$ | ✗ | ✗ | $68.36 \pm 0.20$ | |
| | $\mathcal{A}(\theta_T^{\text{AWS}} \odot m_T^{\text{AWS}})$ | ✔ | ✗ | $67.39 \pm 0.24$ | 0.27 |
| | $\mathcal{A}(\text{abs}(\theta_{\text{init}}) \odot s_T^{\text{LRR}})$ | ✗ | ✔ | $64.92 \pm 0.25$ | |
| | $\mathcal{A}(\text{abs}(\theta_{\text{init}}) \odot s_T^{\text{AWS}})$ | ✔ | ✔ | $\mathbf{67.11 \pm 0.16}$ | |
| | $\mathcal{A}(\theta_T^{\text{LRR}} \odot m_T^{\text{LRR}})$ | ✗ | ✗ | $67.76 \pm 0.17$ | |
| | $\mathcal{A}(\theta_T^{\text{AWS}} \odot m_T^{\text{AWS}})$ | ✔ | ✗ | $66.17 \pm 0.20$ | 0.21 |
| | $\mathcal{A}(\text{abs}(\theta_{\text{init}}) \odot s_T^{\text{LRR}})$ | ✗ | ✔ | $64.03 \pm 0.21$ | |
| | $\mathcal{A}(\text{abs}(\theta_{\text{init}}) \odot s_T^{\text{AWS}})$ | ✔ | ✔ | $\mathbf{66.02 \pm 0.29}$ | |
| | Dense Network | - | - | $59.57 \pm 0.12$ | 1 |
| | $\mathcal{A}(\theta_T^{\text{LRR}} \odot m_T^{\text{LRR}})$ | ✗ | ✗ | $59.18 \pm 0.26$ | |
| | $\mathcal{A}(\theta_T^{\text{AWS}} \odot m_T^{\text{AWS}})$ | ✔ | ✗ | $59.23 \pm 0.18$ | 0.21 |
| | $\mathcal{A}(\text{abs}(\theta_{\text{init}}) \odot s_T^{\text{LRR}})$ | ✗ | ✔ | $57.54 \pm 0.30$ | |
| | $\mathcal{A}(\text{abs}(\theta_{\text{init}}) \odot s_T^{\text{AWS}})$ | ✔ | ✔ | $\mathbf{59.17 \pm 0.19}$ | |
| | $\mathcal{A}(\theta_T^{\text{LRR}} \odot m_T^{\text{LRR}})$ | ✗ | ✗ | $58.95 \pm 0.21$ | |
| | $\mathcal{A}(\theta_T^{\text{AWS}} \odot m_T^{\text{AWS}})$ | ✔ | ✗ | $59.01 \pm 0.21$ | 0.17 |
| | $\mathcal{A}(\text{abs}(\theta_{\text{init}}) \odot s_T^{\text{LRR}})$ | ✗ | ✔ | $57.21 \pm 0.21$ | |
| MLP-Mixer | $\mathcal{A}(\text{abs}(\theta_{\text{init}}) \odot s_T^{\text{AWS}})$ | ✔ | ✔ | $\mathbf{58.93 \pm 0.19}$ | |
| | $\mathcal{A}(\theta_T^{\text{LRR}} \odot m_T^{\text{LRR}})$ | ✗ | ✗ | $58.83 \pm 0.23$ | |
| | $\mathcal{A}(\theta_T^{\text{AWS}} \odot m_T^{\text{AWS}})$ | ✔ | ✗ | $58.79 \pm 0.24$ | 0.14 |
| | $\mathcal{A}(\text{abs}(\theta_{\text{init}}) \odot s_T^{\text{LRR}})$ | ✗ | ✔ | $57.09 \pm 0.19$ | |
| | $\mathcal{A}(\text{abs}(\theta_{\text{init}}) \odot s_T^{\text{AWS}})$ | ✔ | ✔ | $\mathbf{58.73 \pm 0.25}$ | |
| | $\mathcal{A}(\theta_T^{\text{LRR}} \odot m_T^{\text{LRR}})$ | ✗ | ✗ | $58.61 \pm 0.23$ | |
| | $\mathcal{A}(\theta_T^{\text{AWS}} \odot m_T^{\text{AWS}})$ | ✔ | ✗ | $58.77 \pm 0.11$ | 0.11 |
| | $\mathcal{A}(\text{abs}(\theta_{\text{init}}) \odot s_T^{\text{LRR}})$ | ✗ | ✔ | $56.91 \pm 0.21$ | |
| | $\mathcal{A}(\text{abs}(\theta_{\text{init}}) \odot s_T^{\text{AWS}})$ | ✔ | ✔ | $\mathbf{58.62 \pm 0.35}$ | |

