# OpenReview forum: "Find A Winning Sign: Sign Is All We Need to Win the Lottery"
_ICLR.cc/2025/Conference — ICLR 2025 Poster_

### Official Review · Reviewer_q2c9 · 2024-11-01

**Soundness:** 3
**Presentation:** 3
**Contribution:** 3
**Rating:** 8
**Confidence:** 3

**Summary:**

This paper suggests that a signed mask (a binary sparsity mask with sign) is necessary to find the winning tickets in the field of the Lottery Ticket Hypothesis (LTH). Specifically, this paper founds that equipped with the signed mask as well as normalization layer parameters of a subnetwork trained with learning rate rewinding (LRR), a random initialized network can also achieve comparable performance with learning rate rewinding (LRR) trained networks. Extensive experiments on CIFAR-100, Tiny-ImageNet, and ImageNet with various networks demonstrate the effectiveness of signed mask.

**Strengths:**

1. This is a pioneering work that studies the importance of signed mask in the field of LTH. The LTH suggests that the weights initialization, either the original random initialization from the vanilla LTH or some early checkpoints from the weight rewinding, is necessary to win tickets, while this paper suggests that we can rely on the signed mask with random initialization to achieve comparable performance.
From my personal perspective, finding the signed mask is easier in practice than finding the ad-hoc initialization.

**Weaknesses:**

1. This paper only evaluate some lightweight networks on ImageNet such as MobileNet and MLP-Mixer, with an accuracy below 70%. Thus, it's unclear whether the proposed method still generalize well on more powerful networks such as ResNet and Vision Transformers. At least, it would be better if this paper could follow the setting in LRR paper that applies Resnet-50 on ImageNet (see Table 1 of Renda et al., 2020).


2. This paper assumes that networks always include normalization layers and its parameters is necessary, which could limit its generalization. Although normalization is pretty common in modern networks, some early networks don't have normalization layers, such as LeNet and VGG16 without batch normalization [1]. Therefore, it would be better to examine the performance of the signed mask of the LRR subnetwork on these pioneering models to determine if the conclusions hold consistently across different architectures.


3. This paper claims that "an effective signed mask is ALL WE NEED to win the lottery". However, this work still requires the learning rate rewinding to obtains the signed mask with multiple ($T$) iterations. Ideally, the signed mask can be directly obtained from a pre-trained dense networks in the future work.

4. Some mathematical notations seem inconsistent. For example, if I understand correctly, in Line 184, the LRR subnetwork should be $\mathcal{A}( \theta_T^{LRR} ,  m_T^{LRR})$ (same as Line 436), rather than $\theta_0^{LRR}$.  Besides, is $\theta_0$ (Line 172) same as $\theta_{init}$.




[1] https://pytorch.org/vision/main/_modules/torchvision/models/vgg.html#VGG16_Weights

**Questions:**

Please address and discuss the weaknesses above in author response.

---

> ### Author Response · Authors · 2024-11-22
>
> We thank the reviewer for the encouraging comments and constructive feedback.
> We are especially pleased that the reviewer described our work as pioneering and recognized the significance of identifying a winning ticket with any random initialization.
> We have done our best to address the concerns raised, as detailed below.
>
> ## Weakness 1: Experiments on ImageNet
>
> We agree that MobileNetV2 and MLP-Mixer are lightweight models and may not be sufficient to fully demonstrate the generalization of our method.
> As the reviewer suggested, we have evaluated our method using ResNet-50 on the ImageNet dataset.
> Table 1 in the revised manuscript shows that transferring the signed mask obtained using our method to a random initialization is more effective than the case of learning rate rewinding [1] and achieves performance comparable to that of a dense network.
> These results suggest that our conclusion holds even for this larger-scale evaluation.
>
>
>
> Additionally, as the reviewers Kwwx and q2c9 noted, the overall performance of both MobileNetV2 and MLP-Mixer is relatively low in the original manuscript.
> To achieve full convergence, we increased the number of training epochs (from 120 to 300) and we achieved approximately 70% and 60% test accuracy for MobileNetV2 and MLP-Mixer before pruning, respectively, and confirmed that the network had fully converged.
> Using this setup, we reproduced the results in Table 1 and confirmed that the same trends hold true.
> We have revised the results accordingly in the revised manuscript.
>
>
> ## Weakness 2: Dependence on normalization layers
>
> > *This paper assumes that networks always include normalization layers and its parameters is necessary, which could limit its generalization.*
>
> Although there are few deep learning models without normalization layers, investigating several networks without them seems necessary to strengthen the validity of our work.
> As the reviewer suggested, we conducted experiments using LeNet and VGG16 without normalization layers on CIFAR-10 and CIFAR-100 datasets, respectively.
> The results below show that as the sparsity level increases, training a randomly initialized network with the LRR-driven signed mask (third column) leads to performance far below that of the LRR solution (second column) and, in some cases, results in a network that is not trainable (the last row in VGG16 experiments).
> However, we expected that the LRR-driven signed mask could transfer the generalization potential (generalization capacity achievable after training) of the LRR subnetwork since there is no adverse impact of the normalization layer parameters as discussed in Section 3.2.
> We find that these results are worth investigating in our future work.
> Thank you for your suggestion.
>
> **CIFAR-10 with LeNet**
> |Remaining parameter ratio|$\mathcal{A}(\boldsymbol{\theta}_T^\text{LRR} \odot \boldsymbol{m}_T^\text{LRR})$ (%)|$\mathcal{A}(\text{abs}(\boldsymbol{\theta}_\text{init}) \odot \boldsymbol{s}_T^\text{LRR})$ (%)|
> |:---:|:---:|:---:|
> |0.11|65.81|64.94|
> |0.085|65.6|64.63|
> |0.064|65.53|64.32|
> |0.055|64.72|61.58|
> |0.044|64.55|61.52|
> |0.035|63.89|60.83|
>
> **CIFAR-100 with VGG16**
> |Remaining parameter ratio|$\mathcal{A}(\boldsymbol{\theta}_T^\text{LRR} \odot \boldsymbol{m}_T^\text{LRR})$ (%)|$\mathcal{A}(\text{abs}(\boldsymbol{\theta}_\text{init}) \odot \boldsymbol{s}_T^\text{LRR})$ (%)|
> |:---:|:---:|:---:|
> |0.64|69.34|69.21|
> |0.52|69.21|69.11|
> |0.41|69.41|68.94|
> |0.33|69.12|66.13|
> |0.27|69.09|1|
>
>
>
>
>
> ## Weakness 3: Dependence on training multiple iterations
>
> > *However, this work still requires the learning rate rewinding to obtain the signed mask with multiple ($T$) iterations.*
>
> Obtaining a signed mask from a pre-trained model is equivalent to pruning a network in one shot, which significantly damages the network and makes recovering its original performance challenging.
> We agree that iterative pruning is computationally heavy and reducing the cost is a valuable research direction.
> However, iterative pruning is still essential to preserve the original performance, especially when aiming to find a winning ticket that is defined at initialization.
> We would like to emphasize that when we say, 'an effective signed mask is ALL WE NEED to win the lottery,' we mean that the signed mask obtained through our method enables **any** initialized network to perform comparably to a dense network after training.
>
> ## Weakness 4: Several incorrect notations
>
> Thank you for pointing out our mistake.
> We have corrected the typo at Line 184 in the revised manuscript.
> Additionally, $\boldsymbol{\theta}_0$ denotes the initialized parameters at the beginning of iterative magnitude pruning or learning rate rewinding.
>
> This is different from $\boldsymbol{\theta}_{init}$, which refers to arbitrary random initialization.
>
>
> [1] Renda etal, ‘Comparing Rewinding and Fine-tuning in Neural Network Pruning’, in ICLR2020.

---

> ### Comment · Reviewer_q2c9 · 2024-11-26
> **Thanks for your response**
>
> The response clearly solves my concerns. The results on ImageNet with ResNet-50 are promising and verify the generalization of proposed sign mask. Besides, the experiments on networks without normalization layers are interesting. Thus I increase my positive rating.

---

> > ### Author Response · Authors · 2024-11-26
> > **Thanks for raising your rating!**
> >
> > Dear reviewer q2cq,
> >
> > We are glad that our response has adequately addressed your concerns and that you have further increased your positive rating.
> > We believe your feedback has made our paper more convincing and comprehensive.
> > We sincerely appreciate your effort and time.

---

### Official Review · Reviewer_Kwwx · 2024-11-02

**Soundness:** 2
**Presentation:** 2
**Contribution:** 2
**Rating:** 6
**Confidence:** 5

**Summary:**

This paper aims to find lottery ticket hypothesis for neural network models, especially from the perspective of parameter sign. It proposes to slightly change the LRR based on an interpolation method to make the sparse network with good trainability for generalization. Experiments are based on CIFAR and imagenet dataset across different model architectures.

**Strengths:**

1. Find the winning ticket is still an open question in the sparse network training area, such topic is a valuable one to explore.
2. Experimental results shown in the draft support the claimed statement from the paper.

**Weaknesses:**

1. Compared with originla LRR, the change is relatively trivial with not enough research novelty.
2. Used datasets and model architectures are not large-scale like mentioned goal in the abstract. Adding more larger datasets and models will be more supportive.
3. The training configurations for cifar and imagenet may not reach the fully convergence for model training, where 100 and 120 epochs are used for these two, respectively.
4. Overall, the paper is not very clear to read, such as the figures on the main results section. Which line represents the proposed method and which one is for baselines? how to indicate the proposed method outperforms others?
5. In addition, in table.1, more sparsity ratios for experiments will be helpful. And using more other and larger networks will further support the paper claim. More comprehensive empirical studies are needed to support the draft conclusion.

**Questions:**

Please check the weaknesses section above.

---

> ### Author Response · Authors · 2024-11-22
> **Official Comment by Authors (1/2)**
>
> We thank the reviewer for their constructive and detailed feedback.
> We find that addressing the identified weaknesses is essential for improving readability and providing a more convincing evaluation.
> We have addressed all the concerns raised and uploaded a revised manuscript of the paper accordingly.
> We hope the reviewer will carefully consider our feedback and reconsider our work.
>
> ## Weakness 1: Not enough research novelty
>
> > *Compared with original LRR, the change is relatively trivial with not enough research novelty.*
>
> We would like to highlight that the primary contribution of our paper lies not in the proposed techniques but in our observation that **any randomly initialized network** can inherit the generalization potential (generalization capability after training) of the LRR-driven subnetwork through its signed mask and normalization layer parameters.
> However, the reliance on normalization layer parameters presents a hurdle in finding a winning ticket.
> Motivated by the importance of maintaining the basin of attraction of an effective subnetwork to find a winning ticket, we propose a simple yet effective approach that prevents a high error barrier along the linear path connecting the subnetwork trained by AWS (our method) to its counterpart with initialized normalization parameters.
> As a result, we demonstrate that, in relatively large-scale settings compared to those of existing works, AWS successfully identifies an effective signed mask that transfers the generalization potential of the AWS-driven subnetwork when applied to **any randomly initialized network**.
> Compared to recent works either constrained to ad-hoc initialization and small-scale settings or attempting to find a winning ticket in a trained network rather than an initialized one, we believe our work makes significant contributions and will offer valuable insights into the field of the Lottery Ticket Hypothesis.
>
> ## Weakness 2: More larger datasets, models, and sparsity ratio
>
> To further validate our method's generalization in larger-scale settings, we provide the experimental results on ImageNet with ResNet-50 in Table 1 of the revised manuscript.
> Also, we present results on various sparsity levels in Table 2.
> Similarly to the results on MobileNetV2 and MLP-Mixer, we observe that our claim still holds in such a large-scale setting.
>
> We would like to emphasize that iterative magnitude pruning [1], an earlier proposed method, is limited to small-scale settings, such as MNIST with LeNet or CIFAR with 6-layer CNN.
> The follow-up methods, including weight rewinding [2] or learning rate rewinding [3], rely on rewinding to trained weights rather than initialization.
> Thus, finding a winning ticket in relatively large-scale settings remains an open problem.
>
> In this context, we argue that the success of our method with ResNet-50, MobileNetv2, ShuffleNet, EfficientNet, and MLP-Mixer on CIFAR-100, Tiny-ImageNet, and ImageNet datasets extends the lottery ticket hypothesis to relatively larger scales and represents one step forward toward achieving the goal of the lottery ticket hypothesis.
>
>
> ## Weakness 3: Training configurations
>
> To achieve full convergence, we increased the number of training epochs (from 120 to 300 for the ImageNet experiments and from 100 to 150 for both CIFAR-100 and Tiny-ImageNet experiments).
> For the CIFAR-100 and Tiny-ImageNet experiments, we did not observe meaningful changes, but for the ImageNet experiments, we achieved approximately 70% and 60% test accuracy for MobileNetV2 and MLP-Mixer before pruning, respectively, and confirmed that the network had fully converged.
> Using this setup, we reproduced the results in Table 1 and confirmed that the same trends hold true.
> We have revised the results accordingly in Table 1 of the revised manuscript.
> We believe this revision will make our evaluation more convincing. Thank you for pointing out that.
>
> [1] Frankle etal, ‘The Lottery Ticket Hypothesis: Finding Sparse, Trainable Neural Networks’, in ICLR2019.
>
> [2] Frankle etal, ‘Linear Mode Connectivity and the Lottery Ticket Hypothesis’, in ICML2020.
>
> [3] Renda etal, ‘Comparing Rewinding and Fine-tuning in Neural Network Pruning’, in ICLR2020.

---

> ### Author Response · Authors · 2024-11-22
> **Official Comment by Authors (2/2)**
>
> ## Weakness 4: Presentation clarity
>
> > *Overall, the paper is not very clear to read, such as the figures on the main results section. Which line represents the proposed method and which one is for baselines? how to indicate the proposed method outperforms others?*
>
> We agree that Figure 3 is difficult to interpret without reading Section 4.2 and having knowledge of the mathematical notations.
> In the revised manuscript, we clarify what each plot represents in the caption.
>
> > *How to indicate the proposed method outperforms others?*
>
> Figure 3(a) shows that applying the signed mask obtained through LRR to a randomly initialized network (indicated by the orange plots) results in a significant performance drop compared to the results of training with LRR (indicated by the blue plots).
> And it is similar to the performance when the sign information is not used (indicated by the purple plots).
> This implies that the sign information obtained by LRR is not useful to a random initialization.
> On the other hand, applying the signed mask obtained through AWS (our method) to a randomly initialized network (indicated by the red plots) results in performance comparable to that of a network trained with AWS (indicated by the green plots).
> Thus, a randomly initialized network with the AWS-driven signed mask can achieve performance comparable to that of the dense network (indicated by the dotted line).
>
> The SGD-noise instability of a randomly initialized network with the LRR-driven signed mask and the high error barrier along the linear path connecting it to the LRR-driven subnetwork (indicated by the orange plots in Figure 3(b) and (c)) demonstrate that the signed mask obtained through LRR cannot transfer the generalization potential (generalization capability after training) or the basin of attraction of the LRR-driven subnetwork to a randomly initialized network.
> In contrast, a randomly initialized network with the AWS-driven signed mask and the low error barrier along the linear path connecting it to the AWS-driven subnetwork (indicated by the green plots) demonstrate that the AWS-driven signed mask effectively preserves the basin of attraction of the AWS-driven subnetwork even when applied to any random initialization.
>
> These results demonstrate that our method effectively addresses the negative impact of normalization layers, described in Section 3.2, and identifies a singed mask that enables any randomly initialized network to perform comparably to the dense network.

---

> ### Author Response · Authors · 2024-12-02
> **A gentle reminder to Reviewer Kwwx**
>
> Dear Reviewer Kwwx,
>
> Thank you once again for your time and effort in reviewing our work.
> Your valuable comments on novelty, experiments, and presentation clarity have been instrumental in improving our paper.
> We have carefully addressed the concerns raised and submitted a revised manuscript.
> We would greatly appreciate it if you could let us know whether our revisions sufficiently address your concerns.
> Please feel free to share any additional feedback.

---

### Official Review · Reviewer_3AkZ · 2024-11-04

**Soundness:** 3
**Presentation:** 3
**Contribution:** 2
**Rating:** 8
**Confidence:** 4

**Summary:**

The Lottery Ticket Hypothesis (LTH) assumes that there exists a sparse subnetwork inside a dense one that can be trained from initialization to reach a comparable performance to the full model; the aim is to find such a subnetwork, called the winning ticket.
This work focuses around LTH and introduces a novel approach based on a sparse mask with sign information, while considering  a variation of learning rate rewinding (LRR) that encourages linear mode connectivity to bypass issues tied to the initialization of normalization layers. The experimental results vouch for the efficacy of the proposed method; across various architectures and datasets, the proposed sign mask allows for a randomly initialized network to perform comparably to a dense network.

**Strengths:**

The paper is overall well-written and easy to follow. The authors clearly present the motivation of the paper while providing some insights into the proposed approach right from the start.
The related work section is clear giving a good overview of the LTH field and its issues. The proposed LRR variant is simple, considering a random an simple interpolation of the normalization parameters.

The experimental evaluation includes a variety of datasets and architectures with the experimental details clearly stated and the experimental results vouch for the efficacy of the proposed approach.

**Weaknesses:**

I find that the main novelty of the proposed approach is the insights into the normalization issues and the introduced AWS variant. In this context, I do not consider the contribution to be of the highest novelty.

**Questions:**

Why did the authors choose only the MobileNetV2 and MLP-Mixer for the Imagenet setting? Were there any issues with the other architectures?

Can the authors provide a corresponding table (similar to Table 1) for the CIFAR-100 and Tiny-ImageNet experiments?

What are the sparsity levels for these experiments? Do these correspond to the remaining parameter ratio in Fig. 3?

What is the behavior of the proposed approach in different sparsity levels in the case of ImageNet? How does it compare to standard LRR?

---

> ### Author Response · Authors · 2024-11-22
>
> We appreciate the reviewer’s positive comments on the presentation quality, the simplicity of our method, and the experimental results that demonstrate its efficacy.
> The reviewer has expressed concerns regarding the novelty of our work and raised several questions about the experiments.
> We hope the following feedback addresses these concerns and questions.
>
> ## Weakness 1: Novelty
>
> > *I find that the main novelty of the proposed approach is the insights into the normalization issues and the introduced AWS variant. In this context, I do not consider the contribution to be of the highest novelty.*
>
> We would like to highlight that existing works are either constrained to ad-hoc initialization and small-scale settings like iterative magnitude pruning (IMP) [1] or focus on finding sparse networks with trained weights, which do not represent true winning tickets [2,3].
> In contrast, our work is the first to demonstrate that a signed mask, accounting for the effect of normalization layers, enables **any randomly initialized network** to generalize comparably to a dense network, validated at **relatively large scales**.
> We believe that our work is pioneering, as the reviewer q2c9 noted, and will serve as a significant advancement in the field of the lottery ticket hypothesis.
>
> ## Question 1: Issues with the ImageNet experiments
>
> >*Why did the authors choose only the MobileNetV2 and MLP-Mixer for the Imagenet setting? Were there any issues with the other architectures?*
>
> We were initially limited to evaluating our method only on MobileNetV2 and MLP-Mixer due to computing resource constraints.
> However, we have since overcome this issue using cloud computing services and now present additional results on ResNet50, a larger-scale architecture, in Table 1 of the revised manuscript.
>
> ## Question 2: Details on Figure 3
>
> > *Can the authors provide a corresponding table (similar to Table 1) for the CIFAR-100 and Tiny-ImageNet experiments? What are the sparsity levels for these experiments? Do these correspond to the remaining parameter ratio in Fig. 3?*
>
> Please note that Figure 3 provides more information than Table 1, including the performance of LRR/AWS solution and randomly initialized networks with signed masks obtained through LRR/AWS across 15–20 sparsity levels.
> The 'remaining parameter ratio' in Figure 3 corresponds to 1−sparsity.
> In the revised manuscript, we have unified the terminology to avoid confusion by replacing ‘Sparsity’ with ‘Remaining params.’ in Table 1.
> Thanks for your feedback.
>
> ## Question 3: Different sparsity in ImageNet
>
> > *What is the behavior of the proposed approach in different sparsity levels in the case of ImageNet?*
>
> In the revised manuscript, we have added results on different sparsity levels for ImageNet experiments in Table 2. The results demonstrate that across various sparsity levels, training a randomly initialized network with a signed mask obtained through our method outperforms that with a LRR-driven signed mask and performs comparably to the AWS solution.
> Also, a randomly initialized network with a signed mask obtained through our method can achieve performance comparable to the dense network after training.
> These results demonstrate that our claim holds true across various sparsity levels in the ImageNet experiments.
>
> [1] Frankle etal, ‘The Lottery Ticket Hypothesis: Finding Sparse, Trainable Neural Networks’, in ICLR2019.
>
> [2] Frankle etal, ‘Linear Mode Connectivity and the Lottery Ticket Hypothesis’, in ICML2020.
>
> [3] Renda etal, ‘Comparing Rewinding and Fine-tuning in Neural Network Pruning’, in ICLR2020.

---

> > ### Comment · Reviewer_3AkZ · 2024-11-26
> >
> > I thank the authors for their responses which addressed all my concerns, especially with the ResNet experiments. Having also carefully read the concerns and responses to all the reviewers, I decided to increase my score.

---

> ### Author Response · Authors · 2024-11-26
> **Thanks for raising your rating!**
>
> Dear reviewer 3AkZ,
>
> Thank you for carefully reviewing all the comments from other reviewers and our responses, and for deciding to further increase your initial positive rating.
> We were happy to receive your encouraging and constructive feedback and to revise our paper accordingly.
> We sincerely appreciate your effort and time.

---

### Official Review · Reviewer_tJqq · 2024-11-04

**Soundness:** 3
**Presentation:** 3
**Contribution:** 3
**Rating:** 6
**Confidence:** 4

**Summary:**

The paper explores the lottery ticket hypothesis (LTH), which suggests that a sparse network, or "winning ticket," can generalize as well as a dense network when trained from initialization. Previous methods struggled to find such tickets, especially in large-scale settings. The authors propose using a "signed mask," a binary mask with parameter sign information, to transfer generalization potential to a randomly initialized network. They introduce AWS, a variation of learning rate rewinding (LRR) to find A Winning Sign, which addresses the influence of normalization layers and maintains low error barriers. This approach enables a randomly initialized network to perform comparably to a dense network across various architectures and datasets, advancing the search for a true winning ticket in LTH.

**Strengths:**

+ The proposed AWS method leverages a signed mask (a sparse mask with sign information) to transfer the generalization potential of a subnetwork to a randomly initialized network. This approach help addresses a critical challenge in LTH by maintaining the network's basin of attraction, even after initializing all parameters, thus taking a meaningful step towards identifying a true winning ticket.

+ AWS effectively eliminates the reliance on trained normalization layer parameters, which has been a major limitation in previous methods. By encouraging linear mode connectivity between the AWS subnetwork and its counterpart with initialized normalization parameters, the method ensures that the subnetwork remains within its basin of attraction, which avoids the need for parameter retaining.

+ The paper provides empirical evidence demonstrating that the signed mask can enable a randomly initialized network to generalize comparably to a dense network across various architectures and datasets. This shows the method's applicability and effectiveness in various settings.

**Weaknesses:**

- The proposed method heavily relies on the effectiveness of signed mask in preserving the generalization potential of the subnetwork. It is not clear whther the proposed method can be generalized to more model architectures and tasks. If the signed mask does not perform well across different architectures, tasks or datasets, the method's applicability could be limited.

- The method requires careful handling of normalization layer parameters, which adds complexity. The need to interpolate normalization parameters during training may bring challenges when scaling up the proposed method. its scalability to large scale models/datasets is unclear as well.

**Questions:**

Please refer to details of above "Weaknesses" sections.

---

> ### Author Response · Authors · 2024-11-22
>
> We appreciate the reviewer's positive comments on our method and its applicability and effectiveness.
> The reviewer's main concern appears to be the generalization ability and scalability of our method to additional models and tasks.
> We hope to address these concerns through our responses below.
>
> ## Weakness 1: Generalization to more models and tasks
> > *It is not clear whether the proposed method can be generalized to more model architectures and tasks.*
>
> Currently, finding a winning ticket is a challenging and exploratory early-stage research field, where many recent works are still limited to relatively simple models and datasets using the iterative magnitude pruning (IMP) [1].
> For instance, the effectiveness of IMP has been demonstrated in LeNet or 6-layer CNN with MNIST and CIFAR-10 datasets.
> Also, just like other research fields at early stage, the works in this field have performed experiments on classification.
> Although some works deals with other tasks like object recognition [2], they do not focus on a true winning ticket and instead rely on rewinding to trained weights.
>
> In that sense, we want to emphasize that the success of our method with ResNet-50, MobileNetv2, ShuffleNet, EfficientNet, and MLP-Mixer on CIFAR-100, Tiny-ImageNet, and ImageNet datasets extends the lottery ticket hypothesis to relatively larger scales and represents one step forward toward achieving the goal of the lottery ticket hypothesis.
>
> To further validate our method's generalization, we additionally provide the experimental results on ImageNet with ResNet-50, which is a larger-scale experiment, in Table 1 of the revised manuscript.
> Similarly to the results on MobileNetV2 and MLP-Mixer in Table 1, we observe that our claim still holds in such a larger-scale setting.
>
>
>
>
> ## Weakness 2: Scalability of the proposed method
> > *The need to interpolate normalization parameters during training may bring challenges when scaling up the proposed method*
>
>
> If we understand correctly, this concern pertains to the computational overhead introduced by our method. Compared to the original learning rate rewinding [3], our approach simply involves randomly and linearly interpolating normalization parameters with their initialization. Specifically, we randomly sample the interpolation ratio $\alpha$ and obtain $\alpha\cdot\boldsymbol{\psi} + (1-\alpha)\cdot\boldsymbol{\psi}_0$ where $\boldsymbol{\psi}$ and $\boldsymbol{\psi}_0$ indicate trainable and initialized normalization parameters, respectively.
> The interpolated parameters are then used during the forward pass.
> As a result, our method induces almost no computational overhead.
>
>
>
> [1] Frankle etal, ‘The Lottery Ticket Hypothesis: Finding Sparse, Trainable Neural Networks’, in ICLR2019.
>
> [2] Girish etal, ‘The Lottery Ticket Hypothesis for Object Recognition’, in CVPR2021.
>
> [3] Renda etal, ‘Comparing Rewinding and Fine-tuning in Neural Network Pruning’, in ICLR2020.

---

> > ### Comment · Reviewer_tJqq · 2024-11-26
> >
> > I appreciate the auhtors for their responses and they help address some of my concerns. I would like to keep my suggestion for acceptance.

---

> > > ### Author Response · Authors · 2024-11-27
> > > **Thanks for your time and effort!**
> > >
> > > Dear reviewer tJqq,
> > >
> > > Thank you for your positive feedback on our paper.
> > > We believe your comments help us make it more comprehensive.
> > > We sincerely appreciate your time and effort in reviewing our work.

---

### Meta-Review · Area_Chair_vsyW · 2024-12-20

**Metareview:**

This work investigates the lottery ticket hypothesis (LTH), which assumes that there exists a sparse subnetwork (ie, the winning ticket) that can generalize comparably to its dense counterpart after training from initialization. The reviewers acknowledge its several strengths, including the studied problem is important and the proposed method is novel and very well-motivated, and the experimental results effectively demonstrate the validity of the proposed method.

Initially, the reviewers raised concerns regarding the method's generalization ability and scalability, the lack of evaluation on large-scale datasets or models, the technical novelty, and certain details related to the methodology and experimental settings. However, during the rebuttal, the authors successfully addressed most of these issues, leading all reviewers to support the acceptance of the paper. Consequently, the recommendation is to accept the submission.

**Additional Comments On Reviewer Discussion:**

The authors' response effectively addressed most concerns, resulting in reviewers maintaining their positive scores or increasing their ratings toward acceptance.

---

### Decision · Program_Chairs · 2025-01-22

Accept (Poster)